# EPnG: Adaptive Expert Prune-and-Grow for Parameter-Efficient MoE Fine-tuning

## Abstract

Mixture-of-Experts (MoE) architectures have emerged as a scalable backbone for large language models (LLMs), but their adaptation to downstream tasks remains inefficient due to redundant experts and excessive parameter counts. Parameter-efficient fine-tuning (PEFT) methods such as Low-Rank Adaptation (LoRA) reduce training costs, yet they fail to leverage the dynamic routing signals that are intrinsic to MoE. We introduce **EPnG**, an adaptive expert *prune-and-grow* framework for parameter-efficient MoE fine-tuning. EPnG computes expert importance scores during training to identify under-utilized experts for pruning, while reinforcing high-importance experts by expanding their LoRA ranks with orthogonalized initialization. This adaptive loop reallocates limited trainable parameters to the most impactful experts without increasing the overall budget. On OLMoE and Qwen1.5-MoE, EPnG surpasses LoRA under the same parameter budget (+2.1%p and +1.4%p, respectively) on math and code benchmarks, while achieving performance comparable to full fine-tuning with only 0.5–0.7%p of parameters ($\approx 150\times$ fewer). These results underscore the effectiveness of coupling MoE's conditional computation with adaptive PEFT for scalable fine-tuning.

## 1 Introduction

Large Language Models (LLMs) have rapidly advanced the state of natural language processing (NLP) (Grattafiori et al., 2024; Achiam et al., 2023; Liu et al., 2024a). A key factor behind this progress is the Mixture-of-Experts (MoE) architecture, which employs conditional computation by activating only a small subset of experts for each input token (Shazeer et al., 2017; Fedus et al., 2022). This design enables scaling to hundreds of billions of parameters while keeping the per-token computational cost (FLOPs) comparable to that of a much smaller dense model, thereby combining scalability with efficiency. However, these benefits do not directly extend to the fine-tuning stage, as fully fine-tuning MoE models is prohibitively expensive due to the large number of parameters and the need to update all experts (Rajbhandari et al., 2022; Kim et al., 2021; Aminabadi et al., 2022). Efficient fine-tuning of MoE models is particularly important because downstream applications often require domain adaptation or personalization, which cannot be achieved by pretraining alone.

To alleviate the high cost of fine-tuning, research has focused on Parameter-Efficient Fine-Tuning (PEFT) methods (Lester et al., 2021; Li & Liang, 2021). Among them, LoRA (Low-Rank Adaptation) (Hu et al., 2022) introduces a small number of trainable parameters while retaining strong performance, and has become widely adopted. However, most existing PEFT methods were developed with dense architectures in mind and do not sufficiently account for the unique structural characteristics of MoE. For instance, LoRA applies updates uniformly across all modules, whereas prior studies have shown that the relative importance of modules and layers is not uniform (Zhang et al., 2023; Merchant et al., 2020). Such uniform allocation can therefore be particularly inefficient in MoEs, where routing dynamics and expert specialization play a central role.

These shortcomings become even more severe in MoE models. Because only a subset of experts is activated per input, overlooking expert importance can leave critical experts under-adapted while wasting resources on rarely used ones. Moreover, MoEs suffer from expert imbalance, with some experts over-utilized and others rarely activated (Lepikhin et al., 2021; Fedus et al., 2022). Although recent studies have attempted to design MoE-aware PEFT approaches (Wang et al., 2024; Liu et al., 2024b), they either lacked effective solutions or required large numbers of trainable parameters, un-

Figure 1: Overview of EPnG compared with alternative designs: (i) a vanilla MoE layer (left), (ii) MoE with LoRA uniformly applied to all experts (middle), and (iii) our proposed EPnG, where the router selects experts (e.g., experts 2 and $M$) whose LoRA ranks are dynamically expanded ("grow"), while low-importance experts (e.g., experts 1 and 3) are pruned to free capacity.

dermining efficiency. As a result, the fundamental challenge of efficient and stable MoE adaptation remains unresolved, highlighting the need for a PEFT framework that explicitly accounts for MoE's dynamic routing while maintaining parameter efficiency.

Therefore, we propose **EPnG** (Expert Prune-and-Grow LoRA), a novel adaptive PEFT framework designed for efficient MoE fine-tuning. EPnG integrates a prune-and-grow loop into training: (i) compute expert importance scores from router statistics, (ii) prune low-importance experts to release capacity, and (iii) reinforce high-importance experts by dynamically expanding their LoRA rank with orthogonal initialization for stability. Figure 1 illustrates an overview of EPnG). This adaptive mechanism reallocates limited trainable parameters to the most impactful experts, fostering expert specialization and reducing redundancy. Unlike conventional LoRA, which distributes parameters evenly across experts, EPnG continuously aligns resource allocation with evolving routing patterns during fine-tuning.

On average, EPnG achieves consistent improvements over vanilla LoRA with the same number of trainable parameters. Specifically, EPnG yields +2.1 percent point (%p) (+5.8% relative) on OL-MoE and +1.4%p (+2.4% relative) on Qwen1.5-MoE, averaged across math and code benchmarks. Compared to full fine-tuning (FFT), which requires updating the entire model, EPnG achieves competitive performance while tuning over 140× fewer parameters, highlighting its efficiency. This demonstrates the benefit of dynamic expert management and establishes EPnG as a scalable and practical solution for efficient MoE-based LLM fine-tuning.

## 2 BACKGROUND

**Mixture-of-Experts.** Mixture-of-Experts (MoE) architectures have emerged as a key approach for scaling large language models (LLMs) (Grattafiori et al., 2024; Achiam et al., 2023; Liu et al., 2024a). Each MoE layer takes as input a token representation $x \in \mathbb{R}^d$ and routes it to the top-$k$ experts, selected from a set of $M$ experts $\mathcal{E} = \{E_1, \ldots, E_M\}$. The router weight $W_{\text{router}}$ produces logits:

$$h(x) = W_{\text{router}} \cdot x \in \mathbb{R}^M, \tag{1}$$

which are normalized via a softmax distribution over the available $M$ experts. The gate probabilities, obtained by applying a softmax over the router outputs, represent the likelihood of assigning the token to each expert:

$$p_i(x) = \frac{e^{h(x)_i}}{\sum_{j=1}^{M} e^{h(x)_j}}, \quad i = 1, \ldots, M. \tag{2}$$

During forward computation, the top-$k$ experts with the largest gate probabilities are selected:

$$\mathcal{S}(x) = \text{Top-}k\big(p(x)\big), \quad |\mathcal{S}(x)| = k, \tag{3}$$

The final output of the MoE layer is a weighted combination of the selected experts:

$$\text{MoE}(x) = \sum_{i \in \mathcal{S}(x)} p_i(x) \, E_i(x). \tag{4}$$

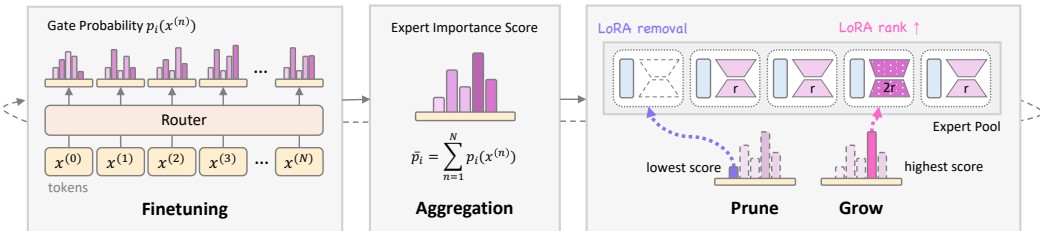

Figure 2: Illustration of EPnG's end-to-end training procedure.

This sparse activation mechanism enables scaling to trillions of parameters without linearly increasing inference cost. However, not all experts are equally utilized or trained in practice. Prior studies have observed significant discrepancies in expert usage and contribution (Shazeer et al., 2017; Chi et al., 2022), highlighting the importance of identifying and managing expert significance. This motivates our approach of quantifying expert importance via router gate probabilities.

**Low-Rank Adaptation.** Low-Rank Adaptation (LoRA) (Hu et al., 2022) enables parameter-efficient fine-tuning by injecting trainable low-rank matrices into pre-trained weight matrices. For a given weight matrix $W \in \mathbb{R}^{d \times d}$, LoRA parameterizes the update as:

$$W' = W + \Delta W, \quad \Delta W = AB, \tag{5}$$

where $A \in \mathbb{R}^{d \times r}$, $B \in \mathbb{R}^{r \times d}$, and $r \ll d$ is the LoRA rank. This reduces the number of additional trainable parameters from $d^2$ to $2dr$, enabling efficient yet effective fine-tuning for large models.

**Challenges in Combining MoE and LoRA.** While LoRA has proven effective in dense transformer models, its direct application to MoE architectures introduces unique challenges. In dense models, every parameter is consistently updated across tokens. In contrast, MoE layers only activate a small subset of experts for each token, leading to sparse and uneven parameter utilization. This discrepancy raises fundamental questions about how to design adaptation strategies that remain both efficient and effective in MoE settings, which motivates our study.

## 3 METHODOLOGY

In this paper, we propose **EPnG** (Expert Prune-and-Grow), a novel adaptive PEFT framework designed for efficient Mixture-of-Experts (MoE). Figure 2 shows its overall procedure. EPnG dynamically reallocates LoRA parameters by continuously monitoring expert utilization throughout finetuning. The process begins by collecting router gate probabilities to estimate how frequently each expert is selected. These are aggregated into expert importance scores, which serve as the basis for adaptively pruning and growing experts. Low-importance experts are pruned by removing their LoRA parameters, releasing budget that is reallocated to frequently used experts by expanding their LoRA ranks. This prune-and-grow loop is repeated during finetuning, allowing EPnG to concentrate capacity on task-relevant experts while eliminating redundancy, all under a fixed parameter budget.

### 3.1 EXPERT IMPORTANCE SCORE AGGREGATION

To enable efficient LoRA allocation in MoE models, we first collect router gate probabilities during finetuning (Figure 2, left). These probabilities reflect how often each expert is selected by the router, providing a direct signal of expert utilization. By aggregating them, we can identify which experts are consistently important and which are underutilized, enabling informed decisions for pruning and rank reallocation.

Specifically, for each token $x^{(n)}$, the router outputs a probability distribution over experts, $\{p_i(x^{(n)})\}_{i=1}^{M}$. We then aggregate these signals across tokens by averaging them (Figure 2, middle), yielding the average gate probability for expert $E_i$:

$$\bar{p}_i = \frac{1}{N} \sum_{n=1}^{N} p_i(x^{(n)}), \tag{6}$$

where $N$ is the total number of processed tokens. We refer to $\bar{p}_i$ as *the Expert Importance Score*, i.e., the average gate probability of expert $E_i$. This aggregation captures how frequently each expert is activated throughout finetuning and provides a principled signal that serves as the foundation for our adaptive prune-and-grow strategy.

## 3.2 PRUNING LOW-IMPORTANCE EXPERTS

Based on the computed importance scores, we identify experts with low utilization for pruning. This process frees up parameter budget that can be reallocated to more influential experts in the growing stage (Section 3.3).

**Expert Selection for Pruning.** To identify underutilized experts for pruning, we compute a threshold based on the aggregated importance scores across all experts in all MoE layers. Specifically, we define the pruning threshold $\tau_p$ as the value below which the lowest $\alpha$ fraction of expert importance scores $\bar{p}_i^{(l)}$ fall, where $\alpha \in (0, 1)$ determines the pruning ratio.

We then select the set of experts to prune as:

$$\mathcal{P} = \{E_i^{(l)} \mid \bar{p}_i^{(l)} \leq \tau_p, \; l \in \{1, \ldots, L\}, \; i \in \{1, \ldots, M\}\} \tag{7}$$

where $E_i^{(l)}$ denotes the $i$-th expert in the $l$-th MoE layer, and $\bar{p}_i^{(l)}$ is its corresponding importance score. $L$ is the number of MoE layers and $M$ is the number of experts in each layer.

Since this thresholding is applied iteratively across cycles, the cumulative fraction of pruned experts after $t$ cycles is given by $1 - (1 - \alpha)^t$, ensuring gradual pruning while preserving the most influential experts. Their parameters are then reallocated to more important experts during the growing phase.

**Pruning Operation.** For each selected expert $E_i^{(l)}$, we remove its LoRA parameters:

$$\Delta W_i^{(l)} = 0, \quad \forall E_i^{(l)} \in \mathcal{P}. \tag{8}$$

This deletion eliminates the contribution of low-importance experts and removes their parameters from the optimizer state, ensuring that pruned experts incur no additional parameter overhead.

This pruning strategy removes the corresponding parameters from the optimizer, freeing up parameter budget that becomes available for reallocation to high-importance experts through the growing operation.

## 3.3 GROWING HIGH-IMPORTANCE EXPERTS

The parameter budget released through pruning is reallocated to the most utilized experts by expanding their LoRA ranks. We adopt a budget-neutral setting, where the number of parameters added during growth matches those removed during pruning. This allows for a controlled comparison with static baselines without increasing the overall parameter count. Although over-budget configurations (e.g., expanding beyond the pruned budget) are possible, we focus on the budget-neutral case for clarity in analysis.

**Expert Selection for Growth.** To identify experts for rank expansion, we compute a global threshold based on the aggregated importance scores $\{\bar{p}_i^{(l)}\}$ across all experts and layers. Specifically, we define the growth threshold $\tau_g$ as the value above which the top $(1 - \beta)$ fraction of scores fall, where $\beta \in (0, 1)$ controls the growth ratio.

The selected set of experts is defined as:

$$\mathcal{G} = \{E_i^{(l)} \mid \bar{p}_i^{(l)} \geq \tau_g, \; l \in \{1, \ldots, L\}, \; i \in \{1, \ldots, M\}\} \tag{9}$$

where $E_i^{(l)}$ denotes the $i$-th expert in the $l$-th MoE layer, and $\bar{p}_i^{(l)}$ is its importance score.

Since this allocation is repeated iteratively, the cumulative fraction of experts receiving expanded ranks after $t$ cycles is given by $1 - (1 - \beta)^t$. This procedure ensures that the most frequently utilized experts progressively gain additional capacity through rank expansion, while preserving budget neutrality with pruning.

**Rank Expansion.** For each expert $E_i^{(l)} \in \mathcal{G}$, we expand its LoRA rank from $r_i^{(l)}$ to $r_i^{(l)} + \Delta r$ by appending new low-rank components. The original LoRA update is $\Delta W_i^{(l)} = A_i^{(l)} B_i^{(l)}$, and after expansion it becomes:

$$\Delta W_i^{(l)\prime} = [A_i^{(l)} \ A_i^{\text{new},(l)}] \begin{bmatrix} B_i^{(l)} \\ B_i^{\text{new},(l)} \end{bmatrix}. \tag{10}$$

This provides greater representational capacity for the most influential experts, ensuring that the additional parameters capture complementary task-specific information rather than redundant updates.

**Initialization and Orthogonalization.** To ensure stable training and avoid redundancy, the newly added parameters are initialized and regularized as follows:

- $A_i^{\text{new},(l)}$ is initialized using Kaiming initialization (He et al., 2015) to preserve activation variance, while $B_i^{\text{new},(l)}$ is initialized to zero, following standard LoRA practice (Hu et al., 2022).

- To encourage diversity in the expanded subspace, $A_i^{\text{new},(l)}$ is orthogonalized with respect to the existing columns of $A_i^{(l)}$:

$$A_i^{\text{new},(l)} \leftarrow (I - Q_i^{(l)} Q_i^{(l)^\top}) A_i^{\text{new},(l)}, \tag{11}$$

where $Q_i^{(l)} = \text{orth}(A_i^{(l)})$ is an orthonormal basis spanning the columns of $A_i^{(l)}$.

This procedure ensures that the newly added directions capture complementary task-specific information rather than duplicating previously learned adaptations, thereby improving the effectiveness of rank expansion.

### 3.4 PRUNE-AND-GROW ADAPTATION LOOP

**Warm-up for Stable Importance Estimation.** Expert importance is not static during finetuning: certain experts that appear important early on may lose relevance later, while initially underutilized experts can become crucial later. To avoid premature pruning decisions when router statistics are still unstable, we use a warm-up stage of $T_w$ steps. During this phase, the model is fine-tuned with LoRA as usual while collecting gate statistics, but no pruning or growth is applied.

**Adaptive Parameter Reallocation.** After the warm-up stage, the prune-and-grow procedure is triggered every $T_p$ steps. At each interval, router statistics $\{\bar{p}_i^{(l)}\}$ are aggregated to evaluate expert importance. Low-importance experts are pruned (Section 3.2), and high-importance experts receive additional rank expansion (Section 3.3). This ensures that the parameter budget is continuously redistributed in response to evolving utilization. Under the constraint $\alpha > \beta$, pruning always releases more parameters than are consumed by growth, guaranteeing that the total number of trainable parameters never exceeds the initial budget.

**Summary.** The prune-and-grow loop thus turns finetuning into a dynamic, budget-aware adaptation process. By reallocating LoRA parameters according to router-derived importance signals, it promotes (i) consistent utilization of the limited parameter budget, (ii) reinforcement of task-relevant experts, and (iii) suppression of redundant adaptations. Algorithm 1 provides the pseudocode for the full training loop, and the overall workflow is illustrated in Figure 2.

## 4 EXPERIMENTS

### 4.1 DATASETS

We evaluate our method across three domains: mathematical reasoning, code generation, and preference-based personalization. For mathematical reasoning, models are fine-tuned on Meta-MathQA (Yu et al., 2024) and evaluated on two widely used benchmarks: GSM8K (Cobbe et al., 2021), which consists of grade-school level word problems, and MATH (Hendrycks et al., 2021),

---

**Algorithm 1:** Expert Prune-and-Grow (EPnG) Training

---

**Input:** MoE-LLM with initial LoRA rank $r_0$, dataset $D$, warm-up steps $T_w$, prune interval $T_p$, pruning and growth ratios $\alpha, \beta$, decay factor $\lambda$

Initialize gate statistics $\bar{p}_i^{(l)} \leftarrow 0$ for all experts

**for** $step = 1$ **to** $max\_steps$ **do**
    Sample batch $x \sim D$ and compute loss
    Update gate probabilities $\bar{p}_i^{(l)}$
    Optimize model parameters via backpropagation
    **if** $step \geq T_w$ **and** $step \bmod T_p = 0$ **then**
        Compute pruning threshold $\tau_p$ and growth threshold $\tau_g$ from $\{\bar{p}_i^{(l)}\}$
        Select experts to prune: $\mathcal{P} = \{E_i^{(l)} \mid \bar{p}_i^{(l)} \leq \tau_p\}$
        Select experts to grow: $\mathcal{G} = \{E_i^{(l)} \mid \bar{p}_i^{(l)} \geq \tau_g\}$
        **foreach** $E_i^{(l)} \in \mathcal{P}$ **do**
            Remove LoRA parameters: $\Delta W_i^{(l)} \leftarrow 0$
        **foreach** $E_i^{(l)} \in \mathcal{G}$ **do**
            Expand LoRA rank: $r_i^{(l)} \leftarrow r_i^{(l)} + \Delta r$
            Initialize $A_i^{\text{new},(l)}$ with Kaiming init, $B_i^{\text{new},(l)} \leftarrow 0$
            Orthogonalize: $A_i^{\text{new},(l)} \leftarrow (I - Q_i^{(l)} Q_i^{(l)\top}) A_i^{\text{new},(l)}$
        Decay gate statistics: $\bar{p}_i^{(l)} \leftarrow \lambda \bar{p}_i^{(l)}$

---

a large-scale dataset covering competition-level mathematics. Performance is measured by Exact Match (EM), i.e., the proportion of predictions that exactly match the ground-truth solution.

For code generation, we finetune on Code Alpaca (Luo et al., 2023), an instruction-tuned dataset for programming tasks. Evaluation is performed on HumanEval (Chen et al., 2021) and MBPP (Austin et al., 2021), both of which are widely adopted for benchmarking code synthesis. We report pass@10 (Chen et al., 2021), which estimates the probability of obtaining a correct solution within ten sampled generations.

Finally, to assess personalization and user alignment, we use PrefEval (Zhao et al., 2025), a recently introduced benchmark designed to measure model quality under preference-based evaluation. In this benchmark, model outputs are pairwise-compared on diverse prompts, and we follow the official protocol by employing GPT-4o-mini (Achiam et al., 2023) as the evaluator to determine win rates against baselines.

## 4.2 EXPERIMENTAL SETTING

We conduct experiments on two publicly available MoE baselines, allenai/OLMoE-1B-7B-0125 (Muennighoff et al., 2025) and Qwen/Qwen1.5-MoE-A2.7B (Qwen, 2024). For comparison, we include several adaptation strategies: Full finetuning (FFT) on dense models, MoE with uniform LoRA allocation, ESFT (Wang et al., 2024), and our proposed EPnG.

LoRA adapters are applied to the `up_proj` and `gate_proj` matrices within transformer blocks. We adopt a prune-and-grow policy with pruning ratio $\alpha = 0.2$ and growth ratio $\beta = 0.2$. The procedure is triggered every $T_p = 50$ steps after an initial warm-up period of $T_w = 100$ steps. To adapt to changing routing patterns during finetuning, aggregated gate statistics are exponentially decayed with factor $\lambda = 0.2$.

For evaluation, we follow the standard protocol of each benchmark. In GSM8K and MATH, we compute exact match accuracy. In HumanEval and MBPP, we report pass@10 by sampling ten candidates per problem. In PrefEval, we adopt GPT-4o-mini as the evaluator for pairwise preference comparison. All evaluation hyperparameters, such as decoding temperature, top-$p$, and maximum generation length, are fixed across methods, and details are provided in the Appendix A.

Table 1: Results on OLMoE after 500 finetuning steps. Metrics: Exact Match (%) for GSM8K/MATH, pass@10 (%) for MBPP/HumanEval, and accuracy (%) for PrefEval. Trainable parameters are reported as the average % of the full model.

| Method | Params. (%) | Math (EM) | | Code (pass@10) | | Personalization | Avg. |
| --- | --- | --- | --- | --- | --- | --- | --- |
| | | GSM8K | MATH | MBPP | HumanEval | PrefEval | |
| FFT | 100 | 64.97 | 24.68 | 43.60 | 25.00 | 34.44 | 38.54 |
| ESFT | 5.08 | 65.13 | 22.49 | 42.20 | 24.39 | 36.11 | 38.46 |
| LoRA (static) | **0.72** | 64.44 | 21.90 | 39.60 | 22.56 | 35.00 | 36.70 |
| Ours (EPnG) | **0.72** | 66.26 | 21.40 | 42.40 | 23.78 | 38.33 | **38.83** |

Table 2: Results on Qwen1.5-MoE after 500 finetuning steps. Metrics: Exact Match (%) for GSM8K/MATH and pass@10 (%) for HumanEval/MBPP. Trainable parameters are reported as the average % of the full model.

| Method | Params. (%) | Math (EM) | | Code (pass@10) | | Avg. |
| --- | --- | --- | --- | --- | --- | --- |
| | | GSM8K | MATH | MBPP | HumanEval | |
| ESFT | 15.91 | 64.14 | 34.54 | 60.40 | 76.83 | **58.98** |
| LoRA (static) | **0.55** | 62.93 | 34.96 | 60.40 | 70.12 | 57.60 |
| Ours (EPnG) | **0.55** | 64.22 | 37.32 | 60.00 | 74.39 | **58.98** |

**Baselines.** We compare our method against the following approaches: *Full finetuning (FFT)* updates all model parameters during adaptation, serving as an upper-bound reference for performance. *LoRA (static)* applies LoRA adapters with fixed rank across all experts, ignoring differences in expert utilization. *ESFT* (Wang et al., 2024) is a recent parameter-efficient finetuning method for MoE models that specializes experts during adaptation. Our implementation builds on the official ESFT codebase. *EPnG (ours)* is the proposed prune-and-grow strategy, which adaptively reallocates LoRA parameters based on expert importance scores.

### 4.3 RESULTS ON OLMoE

Table 1 summarizes the results on OLMoE. Importantly, our method updates only $0.72\%$ of the parameters over $140\times$ smaller than full fine-tuning (FFT), yet still achieves comparable or even better performance. For instance, EPnG slightly outperforms FFT on GSM8K ($66.26\%$ vs. $64.97\%$), while maintaining similar averages across all benchmarks ($38.83\%$ vs. $38.54\%$). Relative to static LoRA with the same parameter budget, EPnG consistently yields improvements, e.g., +1.82 percentage points (%p) on GSM8K and +2.63%p on the overall average.

*Takeaway:* Even under an extremely tight parameter budget, pruning under-utilized experts and reallocating capacity through EPnG preserves or improves performance, demonstrating that full fine-tuning is not necessary to reach strong downstream results.

### 4.4 RESULTS ON QWEN1.5-MOE

Table 2 shows the results on Qwen1.5-MoE, a larger MoE model. Here, ESFT updates $15.9\%$ of the parameters, while EPnG operates with only $0.55\%$, a nearly $29\times$ reduction. Despite this drastic gap in parameter count, EPnG attains the same average performance as ESFT ($58.98\%$), and consistently surpasses static LoRA under the identical budget. Notably, EPnG improves MATH from $34.96\%$ to $37.32\%$ and HumanEval from $70.12\%$ to $74.39\%$, confirming that dynamic allocation of capacity scales effectively to larger MoE backbones.

*Takeaway:* EPnG delivers the efficiency of LoRA with the effectiveness of ESFT, showing that pruning-and-growing can be applied to other MoE models without sacrificing accuracy.

| Method | Acc. |
|--------|------|
| LoRA | 65.50 |
| Grow only | 64.97 |
| Prune only | 65.81 |
| **Prune & Grow** | **66.41** |

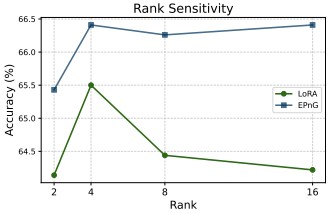 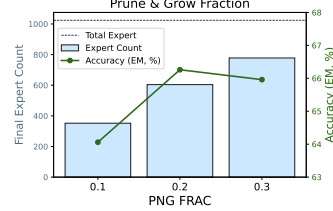

(a) Ablation study.  (b) Rank sensitivity.  (c) Effect of prune-and-grow fraction.

Figure 3: Further analysis of EPnG. (a) Ablation study comparing LoRA, grow-only, prune-only, and prune-and-grow. (b) Sensitivity to LoRA rank, where EPnG maintains stable performance while LoRA fluctuates. (c) Effect of prune-and-grow fraction, showing that moderate pruning achieves the best trade-off between accuracy and expert utilization.

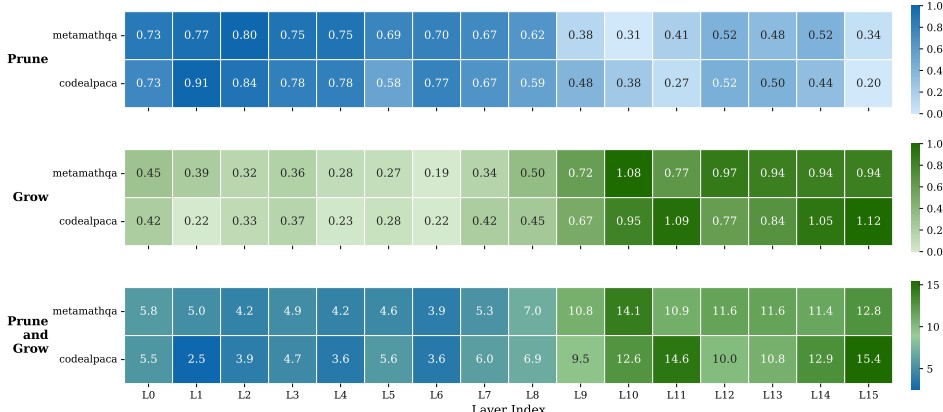

Figure 4: Layer-wise behavior of prune-and-grow. The top heatmap shows pruning ratios, where shallower (earlier) layers are pruned more heavily. The middle heatmap shows growing ratios, highlighting deeper (later) layers where experts are expanded. The bottom heatmap summarizes the resulting average rank distribution: blue indicates reduced ranks, while green indicates expanded ranks. Together, these plots reveal that EPnG adaptively reallocates capacity across layers, focusing resources on the most impactful regions of the model.

## 4.5 OVERALL COMPARISON.

Across both OLMoE and Qwen1.5-MoE, a consistent pattern emerges: EPnG maintains or improves task performance while requiring less than $1\%$ of the trainable parameters. On OLMoE, EPnG matches the accuracy of full fine-tuning with $140\times$ fewer parameters, while on Qwen1.5-MoE, it achieves the same average as ESFT with $29\times$ fewer parameters. These results highlight that pruning-and-growing offers a scalable and parameter-efficient alternative to conventional fine-tuning methods, combining the efficiency of LoRA-style updates with the robustness of more expensive approaches.

These results confirm that EPnG achieves consistent performance gains over static LoRA and MoE-specific ESFT, without exceeding the same parameter budget. By dynamically reallocating capacity toward high-importance experts, EPnG improves both reasoning and generation performance, making it a practical alternative to expensive full finetuning.

## 5 FURTHER ANALYSIS

We conduct additional analyses on gsm8k dataset to better understand why prune-and-grow improves performance under tight parameter budgets. Figures 3 and 4 provide detailed insights.

**Ablation Study.** We conduct an ablation study to examine the contributions of pruning and growing (Figure 3a). Compared to the static LoRA baseline (65.50%), applying either growing (64.97%) or pruning (65.81%) in isolation yields comparable or slightly improved performance. Notably, pruning proves more effective than growing, as it achieves better accuracy with fewer parameters than the baseline. In contrast, combining both operations leads to the best result (66.41%), indicating that pruning and growing complement each other: pruning eliminates redundant components, while growing reinforces useful ones.

We speculate on why "Prune only" outperforms "Grow only." Pruning likely provides an implicit regularization effect, removing redundancy and forcing the model to concentrate on essential representations, which can improve generalization despite reduced capacity. By contrast, growing increases capacity without addressing redundancy and may introduce additional noise, making training less stable. The newly added parameters might also be under-utilized in early training, limiting their contribution. While this interpretation remains tentative, the results suggest that pruning is a particularly effective strategy, achieving better performance than the baseline with fewer parameters.

**Rank Sensitivity.** Figure 3b examines sensitivity to the LoRA rank. LoRA exhibits large fluctuations in accuracy across different ranks, underscoring the difficulty of selecting the right hyperparameter. In contrast, EPnG maintains stable performance, indicating that dynamically reallocating capacity mitigates this sensitivity.

**Prune-and-Grow Fraction.** Figure 3c explores the effect of varying the prune-and-grow fraction. Higher fractions prune a larger set of experts and concentrate capacity on fewer ones. Performance improves until about 0.2, beyond which the benefit plateaus. This suggests that moderate pruning is necessary for the best balance between removing redundancy and preserving expert diversity.

**Layer-Wise Behavior.** Figure 4 visualizes how pruning and growing are distributed across layers. The "Prune" heatmap shows that shallower layers are pruned more heavily, while the "Grow" heatmap highlights deeper layers where experts are expanded. The combined map summarizes the resulting average rank distribution, with blue indicating reduced ranks and green indicating expanded ranks. This demonstrates that EPnG adaptively reallocates capacity toward the most impactful layers, yielding a balanced and efficient expert configuration. For a detailed analysis of how these dynamics evolve across training steps, please refer to Appendix D.

**Summary.** Overall, these analyses show that prune-and-grow not only achieves higher accuracy than its individual components but also stabilizes rank sensitivity, manages pruning trade-offs, and adaptively redistributes capacity across layers. This explains why EPnG consistently surpasses static LoRA under the same parameter budget.

## 6  CONCLUSION

In this work, we introduced **Expert Prune-and-Grow (EPnG)**, a parameter-efficient fine-tuning method for Mixture-of-Experts (MoE) models. EPnG identifies under-utilized experts via gate probabilities and prunes them, while reallocating the released parameter budget to expand the ranks of high-importance experts. This dynamic reallocation enables the model to adaptively shift capacity toward the most impactful components without increasing the overall parameter count.

Extensive experiments on OLMoE and Qwen1.5-MoE demonstrate that EPnG achieves accuracy comparable to or better than full finetuning and ESFT, while updating only about 1% of parameters. Relative to static LoRA, EPnG provides consistent gains across most math, code, and personalization benchmarks, with notable improvements on GSM8K and HumanEval. These results confirm the benefit of combining pruning and growing under a budget-neutral constraint. Our analyses further show that EPnG stabilizes rank sensitivity, balances pruning trade-offs, and adaptively redistributes capacity across layers.

*In summary*, EPnG provides a simple yet effective framework that combines the efficiency of LoRA with the adaptability of dynamic expert reallocation. We believe this approach opens a promising direction for scaling parameter-efficient fine-tuning to even larger MoE backbones and resource-constrained deployment scenarios.

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

# A  IMPLEMENTATION DETAILS

## A.1  BASE MODELS AND BASELINES

We conduct experiments on two publicly available Mixture-of-Experts (MoE) models:

- **OLMoE**: `allenai/OLMoE-1B-7B-0125`
- **Qwen-MoE**: `Qwen/Qwen1.5-MoE-A2.7B`

The following adaptation baselines are included for comparison:

- Full Finetuning (FFT) on dense models
- MoE with uniform LoRA allocation
- ESFT (Wang et al., 2024)
- EPnG (ours)

## A.2  PRUNE-AND-GROW POLICY

- Pruning ratio: $\alpha = 0.2$
- Growth ratio: $\beta = 0.2$
- Trigger interval: $T_p = 50$ steps
- Warm-up period: $T_w = 100$ steps
- Exponential decay factor: $\lambda = 0.2$ (for updating aggregated gate statistics)
- Hard pruning: implemented by zeroing LoRA $B$ matrices during training

## A.3  TRAINING SETUP

- Optimizer: `adamw_torch_fused`
- Learning rate: $1 \times 10^{-4}$
- Batch size: 32
- Training steps: 500
- Precision: mixed precision (bfloat16)
- Hardware: NVIDIA A100 80GB GPU
- rank: 8
- seed: 0

**Remark.**  All experiments are conducted under a single NVIDIA A100 (80GB) GPU setup. Due to memory constraints, full finetuning (FFT) on the Qwen1.5-MoE model could not be performed, and thus we only report FFT results on OLMoE. Other baselines (LoRA, ESFT, and EPnG) are consistently compared across both backbones under the same hardware setting.

## A.4  DATASETS AND EVALUATION PROTOCOL

**Math reasoning.**  Trained on MetaMathQA, evaluated on GSM8K and MATH (restricted to algebra subset). Metric: **Exact Match (EM)**.

**Code generation.**  Trained on CodeAlpaca, evaluated on HumanEval and MBPP. Metric: **pass@10**, using 10 sampled candidates per problem (`temperature=0.7`, `num_return_sequences=10`).

**Personalization.**  Evaluated on PrefEval. Metric: **pairwise preference accuracy**, judged by GPT-4o-mini.

## A.5 Personalization Evaluation Protocol

For personalization evaluation, we adopt the PrefEval benchmark (Zhao et al., 2025), which is designed to test LLMs' ability to infer, memorize, and follow user preferences in conversation. Our implementation closely follows the official protocol: model generations are paired with explicit user preferences, and GPT-4o-mini is employed as an automatic evaluator.

The evaluation covers four dimensions:

- **Acknowledge**: whether the model correctly recognizes the user preference.
- **Violate**: whether the model's response contradicts the preference.
- **Hallucinate**: whether unsupported or fabricated content is introduced.
- **Helpful**: whether the output aligns with the user's preference and provides useful content.

We parse model responses into `<preference>` and `<answer>` tags, construct evaluation prompts with preference, query, and generated response, and obtain judgments via GPT-4o-mini. The resulting structured annotations provide fine-grained error analysis consistent with PrefEval.

All decoding hyperparameters (temperature, top-$p$, maximum generation length) are fixed across models for fairness.

## B Model Configurations

We summarize the key hyperparameters of the MoE models used in our experiments, **OLMoE-1B-7B** (Muennighoff et al., 2025) and **Qwen1.5-MoE-A2.7B** (Qwen, 2024), in Table 3. This provides a clear overview of the architectural differences between the models.

Table 3: A summary of the MoE model configurations used in our experiments.

| Hyperparameter | OLMoE-1B-7B | Qwen1.5-MoE-A2.7B |
|---|---|---|
| Total Parameters | 6.9B | 14.3B |
| Active Parameters | 1.3B | 2.7B |
| Layers ($L$) | 16 | 24 |
| Hidden Dimension ($d_{\mathrm{model}}$) | 2048 | 2048 |
| Number of Experts ($N$) | 64 | 60 + 1 (shared) |
| Top-K Routing ($k$) | 8 | 4 |
| Expert FFN Intermediate Size | 1024 | 1408 |

## C Additional Experimental Results

### C.1 Generability of Fine-Tuned Models

To verify that our method does not compromise the general capability of the backbone, we evaluate on standard reasoning benchmarks without domain-specific finetuning. Table 4 shows that EPnG preserves baseline performance, while full finetuning (FFT) leads to degradation.

| Model | ARC-C | ARC-E | BoolQ |
|---|---|---|---|
| Base | 50.17 | 69.30 | 66.54 |
| FFT | 48.16 | 63.51 | 65.75 |
| LoRA | 50.17 | 68.42 | 65.23 |
| EPnG | 50.17 | 69.12 | 65.75 |

Table 4: General evaluation after finetuning. EPnG maintains generability.

## C.2 TRAINING LOSS COMPARISON

Figure 5 compares training loss curves of EPnG against static LoRA across both backbones. On both Qwen-MoE and OLMoE, EPnG achieves convergence behavior that closely tracks the stability of LoRA. This confirms that dynamic prune-and-grow operations do not introduce training instability, while still enabling expert adaptation during fine-tuning.

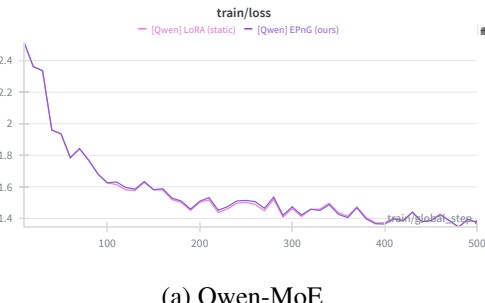
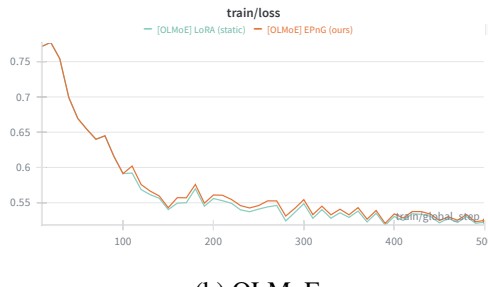

(a) Qwen-MoE                         (b) OLMoE

Figure 5: Training loss comparison between static LoRA and EPnG. EPnG preserves stable convergence while dynamically adapting experts.

## C.3 EFFECT OF LoRA PLACEMENT

We further investigate the effect of attaching LoRA adapters to the `down_proj` matrix in addition to the default `up_proj` and `gate_proj`.

Table 5 shows results on OLMoE with and without the `down_proj` option.

| Method | Params. (%) | GSM8K | MATH | Avg. |
|---|---|---|---|---|
| LoRA (up+gate) | 0.72 | 64.44 | 21.90 | 43.17 |
| LoRA (+down) | 1.09 | 65.35 | 22.33 | 43.84 |
| EPnG (up+gate) | 0.72 | 66.26 | 21.82 | 44.04 |
| EPnG (+down) | 1.09 | 66.19 | 23.50 | 44.85 |

Table 5: Effect of adding `down_proj` to LoRA placement in OLMoE. Adding `down_proj` increases parameters but does not consistently improve accuracy.

# D ANALYSIS OF EXPERT IMPORTANCE DYNAMICS

## D.1 MOTIVATION AND EXPERIMENTAL SETUP

Our training pipeline employs a dynamic *prune-and-grow* strategy to adjust expert ranks during training. To justify this design choice, we first investigate how the distribution of expert importance scores evolves before and after fine-tuning. Rather than fixing the set of active experts statically, we aim to show that adapting the structure based on importance scores is necessary and beneficial. We conducted this experiment by applying a vanilla LoRA adaptation to the mixture-of-experts (MoE) architecture. Specifically, we compared the model at two stages: the base stage before attaching LoRA, and the fine-tuned stage after training with LoRA for 500 steps. The backbone model was `OLMoE-1B-7B-0125`(Muennighoff et al., 2025), and we used the MetaMathQA(Yu et al., 2024) dataset from HuggingFace. From this dataset, we sampled 1,000 examples for routing analysis and statistical evaluation of importance scores.

## D.2 RESULTS

Figure 6 presents the comparison of expert utilization patterns for the base model (top) and the fine-tuned model (bottom). The coloring scheme highlights the top and bottom 20% of experts. Experts

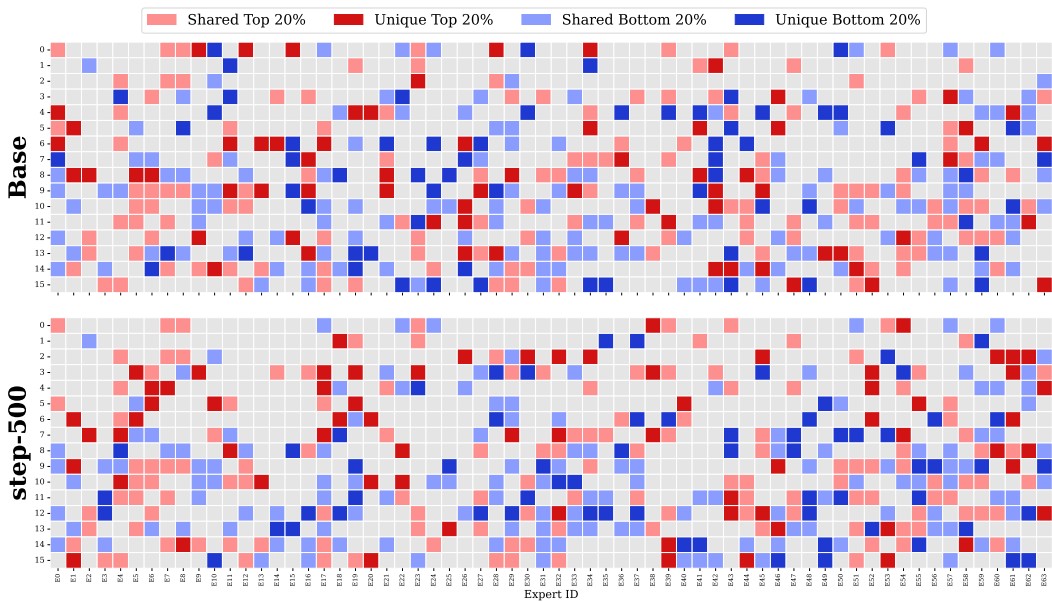

Figure 6: A comparative visualization of expert importance scores. The tables show the expert utilization patterns for the base model (top) and the fine-tuned model (bottom). The coloring scheme identifies the top and bottom 20% of experts, distinguishing between those that are consistently important (**Shared**) and those whose importance changes after fine-tuning (**Unique**).

that remain consistently important across both stages are labeled as **Shared**, while those whose importance changes after fine-tuning are labeled as **Unique**. The analysis reveals that after fine-tuning, the distribution of expert importance shifts significantly, with several experts increasing or decreasing in relevance. This observation provides empirical evidence for the need to adjust expert selection dynamically.

## D.3 DISCUSSION

These findings suggest that LoRA-based fine-tuning not only enables parameter-efficient adaptation, but also induces a meaningful reorganization of expert utilization in MoE models. Consequently, monitoring expert importance and dynamically adjusting ranks through prune-and-grow strategies can lead to more effective parameter allocation and improved model performance.

## E LLM USAGE

We used an LLM to refine the sentences and ensure grammatical accuracy.

