# OpenReview forum: "EPnG: Adaptive Expert Prune-and-Grow for Parameter-Efficient MoE Fine-tuning"
_ICLR.cc/2026/Conference — ICLR 2026 Conference Withdrawn Submission_

### Official Review · Reviewer_Fmsf · 2025-10-29

**Soundness:** 2
**Presentation:** 2
**Contribution:** 2
**Rating:** 2
**Confidence:** 4

**Summary:**

This paper presents a novel Parameter-Efficient Fine-Tuning (PEFT) framework, termed EPnG (Expert Prune-and-Grow), for Mixture-of-Experts (MoE) models. The proposed framework dynamically adapts to routing patterns to enhance fine-tuning performance under a fixed parameter budget. EPnG integrates a prune-and-grow loop into the training process by: (i) computing expert importance scores based on router statistics; (ii) pruning low-importance experts to free up capacity; and (iii) growing high-importance experts through dynamic expansion of their LoRA ranks with orthogonal initialization to ensure stability. This adaptive mechanism reallocates limited trainable parameters to the most impactful experts, thereby enhancing both efficiency and effectiveness without increasing the overall parameter count.

**Strengths:**

- Adaptive and Stable Training Mechanism: The paper introduces a prune-and-grow adaptation loop, comprising a Warm-up and an Adaptive Parameter Reallocation. The Warm-up prevents premature pruning or growth decisions when routing patterns are unstable during early training. A notable feature is the incorporation of a decay factor (λ) in the reallocation, which reduces the influence of older historical information. This design effectively phases out experts that are initially important but become redundant later, while reinforcing newly emergent critical experts.

- Rigorous Budget-Neutral Design: The proposed "budget-neutral" design ensures that the total number of trainable parameters never exceeds the initial budget by enforcing that parameters released through pruning are greater than or equal to those consumed by growth (α > β). This is a crucial feature for resource-sensitive applications.

**Weaknesses:**

Incomplete Literature Review: This paper lacks a discussion of related work in the MoE and PEFT. For instance, "Pushing Mixture of Experts to the Limit: Extremely Parameter Efficient MoE for Instruction Tuning" [ICLR 2024] has explored similar themes. It's recommended that the authors supplement and contextualize their contributions by reviewing and comparing their method with existing works, which would better demonstrate the novelty and positioning of EPnG.

Lack of Result Analysis: In Table 1, EPnG's performance on GSM8K is lower than that of FFT. Please analyze the reasons for EPnG's underperformance on GSM8K and other similar cases.

Non-standard and Unclear Figures and Tables:
 - Inconsistent Visualization Schemes: The color schemes and interpretations across the three heatmaps in Figure 4 are inconsistent and confusing. The top heatmap uses light colors to highlight key points, while the middle one uses dark colors. Please adopt a unified and clear color scheme with a consistent legend to improve the figure's interpretability.
 - Missing Metric Interpretation: The evaluation metrics in Tables 1 and 2 (e.g., Exact Match, pass@10) are presented without indicating whether higher or lower values are better. Please consider adding a brief note for each metric to significantly enhance readability.
 - Unclear Calculation of Average Scores: The average score for the "Ours" method in Table 1 is reported as 38.83. However, the direct arithmetic mean of the provided values (GSM8K, MATH, MBPP, HumanEval, PrefEval) calculates to approximately 38.43, which does not match the reported value. In contrast, the average in Table 2 is clearly a direct mean. It is therefore recommended that the calculation method for the average in Table 1 be explicitly clarified.

Insufficient Description of Experimental Setup:
 - Underspecified Dataset for Personalization Evaluation: In Table 1, the evaluation across three domains fails to specify the dataset used for the "preference-based personalization" benchmark, whereas datasets are clearly named for the "math" and "code" domains. Please explicitly state the dataset used for this evaluation to ensure completeness and reproducibility.
 - Missing Experimental Conditions: Figure 3c, which analyzes the effect of the prune-and-grow fraction, lacks a description of the underlying experimental conditions, such as the base model and the dataset used for the test. It's recommended to add these essential details to the figure caption or main text to facilitate correct interpretation of the results.

**Questions:**

1. Beyond this performance improvement, what new conceptual insight or principle does EPnG reveal about the fine-tuning dynamics of MoE models?

2. The method involves several key hyperparameters (e.g., T_w, T_p, α/β). How sensitive is the performance to these choices? Have you conducted any experiments to demonstrate that the same set of hyperparameters can generalize robustly across different model architectures (e.g., OLMoE vs. Qwen) or task domains?

3. The current "budget-neutral" design is a safe choice for a fair comparison. However, a more exciting capability of an adaptive system would be to intelligently request more parameters for critically important experts. Did you explore any "over-budget" scenarios? If so, what were the results? If not, what is your perspective on the potential and challenges of such a non-budget-neutral paradigm?

---

### Official Review · Reviewer_ztjJ · 2025-10-31

**Soundness:** 2
**Presentation:** 2
**Contribution:** 2
**Rating:** 4
**Confidence:** 4

**Summary:**

This paper proposes EPnG, an adaptive and simple expert prune-and-grow framework for parameter-efficient Mixture-of-Experts (MoE) fine-tuning. It does so by (i) tracking each expert’s gate probabilities to score importance, (ii) hard-pruning the least-used experts’ LoRA adapters, and (iii) growing capacity by expanding the LoRA rank only for the most-used experts. This prune-and-grow cycle begins after a warm-up phase and repeats every fixed number of steps, within the budget of the initial LoRA parameters.

Experiments were conducted on two MoE baselines (OLMoE and Qwen1.5-MoE) across three domains (mathematical reasoning, code generation, preference-based personalization) and show similar or slightly better accuracy than three previous strategies (FFT, ESFT, static LoRA) while using 99% fewer parameters than FFT, 95% fewer than ESFT, and the same number of parameters as static LoRA.

**Strengths:**

- Empirical gains are visible under the same trainable-parameter budgets.

    EPnG beats static LoRA on both OLMoE and Qwen1.5-MoE (≈+2.1 and +1.4 percentage points averaged over math/code).

- Extreme parameter efficiency versus full fine-tuning.

    It achieves accuracy competitive with full fine-tuning while modifying only ~0.5–0.7% of parameters (≈140× fewer), which is a strong practical efficiency claim.

- Practical and simple methodology.

    It uses the router’s gate probabilities to score experts, then prunes low-importance experts’ LoRA adapters and grows rank on high-importance ones, which is efficient and aligns with how MoE routes tokens.

**Weaknesses:**

- Experimental setup is too specialized and does not provide a general and fair assessment of the framework.

    The short, 500-step fine-tuning setup is prone to noise, and early stopping can disproportionately favor adaptive methods.

    Using a single seed (seed = 0) fails to capture run-to-run variance and makes small accuracy deltas ambiguous.

    Removing FFT on Qwen due to memory limits may hide an upper bound and makes efficiency claims rely solely on ESFT, which already operates under a trainable-parameter budget. Therefore, the contrast is less informative.

    The MATH evaluation is limited to the algebra subset rather than the full benchmark.

- Novelty is questionable.

    Similar ideas of dynamically reallocating a LoRA budget via pruning and growing appear in AdaLoRA. The contribution here is largely an engineering choice that leverages the router’s probabilities; it does not introduce a new concept or analytical perspective.

    The primary novelty is the integration of dynamic-reallocation PEFT with MoE, which reads as a domain-specific combination of established techniques rather than an algorithmic breakthrough.

- The motivation is not firm.

    The paper claims that “MoEs suffer from expert imbalance” but EPnG does not directly address this problem, nor does it provide quantitative evidence in the experiments.

    Section 1, paragraph 2 is confusing in its discussion of the “shortcomings of PEFT for MoE” and weakens the motivation.

**Questions:**

- You state the budget never exceeds the initial one if α > β (Sec. 3.4), but experiments fix α = β = 0.2. Please reconcile this with the “never exceeds” claim and provide the exact parameter-count ledger per prune-and-grow cycle.
- Sec. 3.2 says you delete LoRA params and remove optimizer state; Appendix A.2 says “hard pruning” is implemented by zeroing B during training. Which approach is used in the reported numbers, and does optimizer state persist?
- All runs use seed = 0 for 500 steps. Provide 3–5 seeds with means ± 95% CIs for all tables; justify 500 steps and show whether conclusions hold at 1k–2k steps.
- You omit FFT on Qwen1.5-MoE due to memory. Give detailed GPU-memory traces for FFT/ESFT/EPnG, and consider a smaller dense surrogate to keep a true upper bound.
- Provide the exact evaluator prompts, tie-breaking policy, and any system prompts used for PrefEval with GPT-4o-mini, and consider adding an ablation with a second judge (e.g., deterministic rubric) to test robustness.

---

### Official Review · Reviewer_CUUd · 2025-10-31

**Soundness:** 3
**Presentation:** 3
**Contribution:** 3
**Rating:** 6
**Confidence:** 2

**Summary:**

This paper targets the inefficiency of applying standard parameter-efficient fine-tuning (PEFT) methods, such as LoRA, to Mixture-of-Experts (MoE) architectures. The authors argue that LoRA allocates parameters uniformly across experts, neglecting the dynamic routing and varying significance of experts in MoE models.
To address this, the paper proposes EPnG (Expert Prune-and-Grow)—an adaptive framework that periodically prunes low-importance experts (measured via router gate probabilities) and reallocates the released budget by expanding the LoRA rank of high-importance experts, under a fixed total parameter constraint. This process is repeated in a prune-and-grow loop, guided by router statistics.
Experiments on OLMoE and Qwen1.5-MoE show that EPnG achieves accuracy comparable to full fine-tuning while updating less than 1% of parameters, outperforming both static LoRA and prior MoE-specific PEFT baselines (e.g., ESFT). The paper further provides ablations, sensitivity analyses, and visualizations to interpret pruning/growing behavior across layers.

**Strengths:**

（1）Novel and Pragmatic Framework: EPnG introduces an adaptive pruning and growing budget-neutral mechanism, that uses explicit router statistics to guide LoRA rank reallocation. The approach bridges the gap between MoE routing dynamics and PEFT’s efficiency goals, providing a practical solution without requiring architectural modifications.
（2）Strong Empirical Results: On two open MoE backbones (OLMoE, Qwen1.5-MoE), EPnG consistently matches or surpasses full fine-tuning performance while training only 0.5–0.7% of parameters. The results demonstrate both parameter efficiency (≈140× smaller) and robust accuracy across math, code, and personalization benchmarks.
（3）Comprehensive Methodological Design and Analysis: Includes detailed algorithmic description with explicit pruning/growing criteria, orthogonal initialization for stability, and warm-up scheduling for balanced adaptation.
Ablations (prune-only, grow-only, combined) and sensitivity studies provide clear evidence of each component’s impact. Layer-wise analysis visualizes how EPnG redistributes capacity toward deeper layers and frequently utilized experts.

**Weaknesses:**

（1）Limited Theoretical Insight into Pruning/Growing Policy: The pruning threshold and growth ratios (α, β) are empirically chosen without theoretical justification. Suggestion: Introduce or discuss an adaptive control mechanism (e.g., based on parameter utility uncertainty or reinforcement signals) to learn α, β dynamically.
（2）Restricted Evaluation Scope: Experiments use only 500 fine-tuning steps and limited datasets; the long-term stability of continual pruning/growing remains untested. Suggestion: Evaluate on longer training runs or more domain-diverse tasks to verify general effectiveness and adaptability.
（3）Lack of Cost and Efficiency Analysis: While the method improves parameter efficiency, runtime and memory costs are not discussed. Suggestion: Provide wall-clock runtime comparisons and per-step memory consumption to quantify the real-world benefits.

**Questions:**

（1）Dynamic Ratios: What is the empirical sensitivity of performance to pruning and growth ratios (α, β)? Could these be adaptively tuned during training to better reflect evolving expert distributions? Since expert importance depends on router gate probabilities, how stable or noisy are these measurements early in fine-tuning? Could stochastic routing lead to premature pruning of potentially useful experts?
（2）Budget Neutrality Constraints: Does the enforced parameter neutrality limit potential performance improvements in highly sparse settings? Would relaxing this constraint significantly improve results?
（3）Computation and Scalability: How does EPnG scale with deeper MoE layers or larger numbers of experts? Are the compute and memory overheads manageable on commodity hardware?
（4）Generalization Across Domains: The results focus on math, code and personalization tasks. Would the same adaptive mechanism perform well on broader LLM tasks such as dialogue, reasoning or summarization?

---

### Note · Authors · 2025-11-13

I have read and agree with the venue's withdrawal policy on behalf of myself and my co-authors.